# Optimizing management of low back pain through the pain and disability drivers management model: A feasibility trial

Christian Longtin[1], Simon Décary[2‡], Chad E. Cook[3‡], Marc O. Martel[4‡], Sylvie Lafrenaye[5‡], Lisa C. Carlesso[6‡], Florian Naye[1‡], Yannick Tousignant-Laflamme[1,7]*

1 School of Rehabilitation, University of Shebrooke, Sherbrooke, Quebec, Canada, 2 Faculty of Medecine, Laval University, Quebec, Quebec, Canada, 3 Department of Orthopaedics, Duke University, Durham, North Carolina, United States of America, 4 Faculty of Dentistry & Departmet of Anesthesia, McGill University, Montreal, Quebec, Canada, 5 Faculty of Medecine, University of Sherbrooke, Quebec, Quebec, Canada, 6 School of Rehabilitation Science, Faculty of Health Sciences, McMaster University, Hamilton, Ontario, Canada, 7 Research Centre of the Centre Hospitalier Universitaire de Sherbrooke, Centre Intégré Universitaire de Santé et Services Sociaux de l'Estrie, Sherbrooke, Quebec, Canada

☉ These authors contributed equally to this work.
‡ These authors also contributed equally to this work.
* Yannick.Tousignant-Laflamme@usherbrooke.ca

**Data Availability Statement:** All relevant data are within the manuscript and its Supporting information files.

## Abstract

### Introduction

Self-reported levels of disability in individuals with low back pain (LBP) have not improved in the last decade. A broader perspective and a more comprehensive management framework may improve disability outcomes. We recently developed and validated the Low Back Pain and Disability Drivers Management (PDDM) model, which aims to identify the domains driving pain and disability to guide clinical decisions. The objectives of this study were to determine the applicability of the PDDM model to a LBP population and the feasibility of conducting a pragmatic trial, as well as to explore clinicians' perceived acceptability of the PDDM model's use in clinical settings.

### Methods

This study was an one-arm prospective feasibility trial. Participants included physiotherapists working with a population suffering from LBP and their patients aged 18 years or older presenting with a primary complaint of LBP that sought a new referral and deemed fit for rehabilitation from private and public clinical settings. Clinicians participated in a one-day workshop on the integration of the PDDM model into their clinical practice, and were asked to report various LBP-related outcomes via self-reported questionnaires (i.e., impact of pain on physical function, nervous system dysfunctions, cognitive-emotional factors, work disabilities) at baseline and at six-week follow-up. Physiotherapists' acceptability of the use of the PDDM model and appreciation of the training were assessed via semi-structured phone interviews. Analyses focused on a description of the model's applicability to a LBP population, feasibility outcomes and acceptability measures.

**Funding:** YTL received funding (grant) from the Quebec Research Pain Network (QPRN) for a grant of 25 000$ CAD (https://qprn.ca/en/). The funders had no role in study design, data collection and analysis, decision to publish, or preparation of the manuscript.

**Competing interests:** The authors have declared that no competing interests exist.

## Results

Applicablity of the PDDM model was confirmed since it successfully established the profile of patients according to the elements of each categories, and each of the 5 domains of the model was represented among the study sample. Trial was deemed feasible contingent upon few modifications as our predefined success criteria for the feasibility outcomes were met but feasibility issues pertaining to data collection were highlighted. Twenty-four (24) clinicians and 61 patients were recruited within the study's timeframe. Patient's attrition rate (29%) and clinicians' compliance to the study protocol were adequate. Clinicians' perceived acceptability of the use of the model in clinical settings and their appreciation of the training and online resources were both positive. Recommendations to improve the model's integration in clinical practice, content of the workshop and feasibility of data collection methods were identified for future studies. A positive effect for all patients' reported outcome measures were also observed. All outcome measures except for the PainDetect questionnaire showed a statistically significant reduction post-intervention ($p < 0.05$).

## Conclusion

These findings provide preliminary evidence of the potential of the PDDM model to optimize LBP management as well as conducting a future larger-scale pragmatic trial to determine its effectiveness.

## Trial registration

Clinicaltrial.gov: NCT03949179.

## Introduction

Low back pain (LBP) is highly prevalent (up to 84% lifetime prevalence), recurrent [1] and is the leading cause of disability in high- and low-income countries in terms of years lived with disability [2]. Despite increased awareness to improve health systems and for decision makers to improve care, self-reported levels of disability in individuals with LBP have not improved in the last decade [3]. Managing ongoing disability is crucial since it is a strong predictor of chronicity [4] and has significant negative societal impacts and high costs [5]. Factors involved in the development of persistent LBP-related disability include psychological, biological, social, and environmental influences [4, 6].

In order to more effectively manage LBP through rehabilitation approaches, evidence-based publications suggest that rehabilitation professionals, such as physiotherapists (PTs), should use classification systems to frame their diagnosis and guide treatments [7]. Yet, these tools mainly focus on addressing deficits related to biological aspects explaining pain and disability and poorly integrate contextual factors driving the experience of pain, such as the person's environment (social, physical) and inherent personal factors [8]. The use of classification systems based exclusively on biological aspects alone can lead to an incomplete clinical profile and inadequate specific relevant interventions (i.e., providing core stability exercises alone to a patient presenting high levels of cognitive-affective factors), which ultimately marginalize clinical outcomes [8, 9]. Hence, a more comprehensive and broader perspective that can integrate a "true" biopsychosocial framework is needed.

With that in mind, we recently proposed a non-pharmacological management model—the Low Back Pain and Disability Drivers Management (PDDM) model that encompasses the multidimensional elements included within the International Classification of Functioning, Disability and Health (ICF) framework [10, 11]. Detailed description of the model is presented elsewhere [12] and is summarized briefly later in this article. This newly developed model has shown sufficient face and content validity following a modified Delphi survey with a group of experts in musculoskeletal (MSK) pain management [13]. Our model consists of an evaluation framework that builds upon decades of LBP research to improve clinical decision-making processes. It aims to identify the domains influencing pain and disability to create an ICF-based profile or phenotype, which could help rehabilitation clinicians provide more targeted care, optimize treatment outcomes and resource utilization in the management of LBP.

Our long term goal is the integration of the PDDM model into clinical practice management. In the context of this study, our specific objectives were to:

1. Determine the applicability of the PDDM model in a LBP population and the feasibility of participants recruitment, retention rate, suitability of eligibility criteria and compliance to study protocol;

2. Explore clinicians' perceived acceptability of the PDDM model's use in clinical practice;

3. Explore the model's short-term effects (i.e., 6 weeks) on patient-reported clinical outcomes (i.e., levels of pain and disability).

## Methods

### Design

A prospective feasibility trial with a cohort of clinicians and patients was carried out in two clinical settings. The CONSORT statement extension for randomized pilot and feasibility trials [14] and the Transparent Reporting of Evaluations with Nonrandomized Designs (TREND) statement [15] guided the development of this study (S1 and S2 Tables). This is the first step in a research process prior to determine the effectiveness of the PDDM model in the management of LBP. This design has been chosen to gather pilot data on applicability, feasibility and acceptability outcomes as well as the model's preliminary effect in real clinical settings effect through both quantitative and qualitative methods. The applicability of the model refers to its ability to capture each patient's phenotype or clinical profile. Feasibility includes issues such as willingness of individuals to participate in the study, the adherence/compliance of participants to the intervention and whether the intervention can be delivered as intended within the clinical setting [16]. Finally, acceptability is defined as a "multi-faceted construct that reflects the extent to which people delivering or receiving a healthcare intervention consider it to be appropriate" [17]. It includes several constructs such as attitude towards the intervention, the intervention's burden for the individual, the intervention coherence, perceived effectiveness and self-efficacy in performing the intervention. The authors confirm that all ongoing and related trials for this intervention are registered. However, we encountered some technical difficulties during the trial registration process, which explains why the registry entry suggests that this study was registered after patient recruitment began, which is not the case.

### Study setting

The study took place in two different physiotherapy clinical settings: physiotherapy outpatient clinics within a hospital-based setting in the *Centre Intégré Universaitre de Santé et Services*

*Sociaux de l'Estrie (CIUSSSE)* (urban, semi-urban and rural settings) and private physiotherapy clinics from the network of clinics of *PhysioExtra* in the greater region of Montréal, Quebec (urban setting). These clinical settings had expressed interests in participating and we established collaborations with the clinical managers from the two sites from past research projects. Both settings had easy access to patients suffering from MSK disorders, such as LBP. These two settings represent both the private and public sectors of the healthcare system in Canada (Québec).

## Participants

To be included, clinicians from either setting had to: 1) be working with a population suffering from LBP; 2) be able to participate in a one-day training workshop (intervention); 3) assess and initiate treatment of their patients presenting with LBP guided by the PDDM model and; 4) understand French or English. Patients had to be 18 years or older with a primary complaint of LBP, defined as pain primarily in the low back region but sometimes referring to the lower limb [18], and willing to provide patient related-outcomes following their assessment and treatment in physiotherapy. Only new patients presenting with LBP were eligible to participate. Patients not deemed fit for rehabilitation by their therapist following the initial assessment were excluded; the main criterion was the presence of red flags or serious spinal pathology (i.e., cancer, infection, fracture) that would refrain the patient from participating in rehabilitation. In that case, they were referred to the appropriate health professional by their clinician according to standard practice.

## Recruitment

We used a convenience sampling method to recruit both the clinics and the clinicians. Clinic managers of the participating clinical settings provided all the relevant study details and its implication to all their clinicians in each setting. Coordinates of the research team were given to each clinician if interrogations persisted. Clinicians that expressed interest and met the eligibility criteria were enrolled in the study. Informed written consent was provided during the training workshop. Patients who participated in the study were recruited based on the participating PTs' caseload, and their eligibility was assessed by their therapist during the initial visit. Recruitment and data collection took place between May 2019 and December 2019. Each PT was asked to recruit participants by explaining in detail the purposes, process and implications of participating in the study. The patient gave informed consent online via a survey on the LimeSurvey® platform (see next section). The study protocol was approved by the Ethics Review Board of the *CIUSSSE—Centre Hospitalier Universitaire de Sherbrooke (CHUS)* (project number: MP-31-2019-3131) on March 26, 2019. The first patient was recruited on June 4, 2019 and the last patient follow-up was on December 17, 2019.

## Intervention

Recruited clinicians underwent training specific to the model, enabling them to deliver the intervention to their patients based on the principles of the PDDM model [12, 13]. The objectives of the workshop were to provide to the clinicians the knowledge and skills to:

1. Understand the theoretical foundations underpinning each of the five domains of the PDDM model (nociceptive pain drivers, nervous system dysfunction drivers (NSD), comorbidity drivers, cognitive-emotional drivers and contextual drivers) and the specific elements related to each domains;

2. Utilize the different measurement tools to assess the contribution of each domain;

3. Establish the profile of the patient by assessing and identifying which drivers are contributing to the clinical picture, and;

4. Integrate assessment findings for the selection of appropriate interventions to address problematic domains (S1 Fig).

The 1-day workshop was provided by the lead investigator (YTL) with support from co-investigators at the participating clinics and its content included a presentation of the model to facilitate the integration and operationalization of the different concepts (part 1) followed by the demonstration of the different measurements tools specific to the PDDM model and the exploration of different intervention strategies based on the clinical profile of the patient supported by the presentation of two case studies (part 2). Clinicians' integration of the PDDM model into their practice was also supported by the development of a website to facilitate the use of online resources (e.g., short educational capsule, treatment algorithms, and links to questionnaires– www.pddmmodel.wordpress.com).

In addition to the online resources, the model's clinical application was facilitated by the use of an electronic tool hosted on LimeSurvey® and developed by the research team. This tool facilitated the identification by the clinicians of the relative contribution of each domain to the patient health condition by creating a complete personalized profile based on 1) the results of self-reported outcomes measures relevant to the model's domains and 2) clinician's responses to specific questions following their assessment (S2 Fig). The tool was presented in detail to the clinicians during the workshop.

### Outcomes and measures

**Sociodemographic measures.** Individual characteristics including age, sex, clinical setting, education level and years of work experience of the participating clinicians were obtained using self-administered questionnaires handed out during the training workshops (T0). For patients, age, sex and duration of back pain were collected by the clinicians via questionnaires embedded in the electronic data collection tool. This information allowed them to complete their patient's assessment with the use of the model. All patients' related data were collected via the LimeSurvey® platform at the initial visit (T2) and 6 weeks later (T3). The detailed timeline of the data collection for this study is presented in S3 Fig.

**Applicability outcomes.** The primary outcomes for this trial were descriptive, focusing on the applicability of the model in a LBP population, feasibility of the trial design and accep ability of the use of the PDDM model. To determine the applicability of the model, the percentage of patients that fell into each category of the five domains of the model was determined to explore all possible profile of patients, based on the PDDM, which informed on the model's capability to categorize each patient according to their clinical profile. For each domain, four options were possible:

- (A) Presence of at least one element of the Category A,

- (B) Presence of at least one element of the Category B,

- (A+B) Presence of at least one element of the Categories A and B

- (0) Absence of elements in either A or B categories.

The categories are not mutually exclusive so as to account for the heterogeneity of clinical presentations. Detailed description of the categories and elements required to establish the profile according to each domain is presented in S3 Table.

**Feasibility outcomes.** For this trial, the feasibility outcomes included:

1. Feasibility of recruitment: we aimed to recruit a minimum of 10–15 clinicians and 30–45 patients overall within the allocated timeframe;

2. Retention rate and attrition: % of enrolled patients (T2) who reported data at follow-up (T3). Attrition of 30% or lower would be considered indicative of a successful feasibility trial with a plan of repeating the retention strategies for the full trial [19];

3. Suitability of eligibility criteria was determined based on overall recruitment (i.e., feasibility of recruitment) and clinicians' feedback based on two questions: are the criteria sufficient or too restrictive? Is it obvious who meets and who does not meet the eligibility criteria? [20];

4. Clinicians' compliance to the study protocol was assessed by the reporting of the patients' clinical data by the clinicians according to the PDDM model (at T2 and T3) and their participation in both semi-structured interviews at T1 and T3. We aimed for an overall compliance rate of >80% among all clinicians;

**Acceptability outcomes.** The PTs' acceptability outcomes were assessed via semi-structured phone interviews and included:

1. PTs' appreciation of the workshop (assessed within a week after the training, T1), the web-based support resources and the data collection tool (assessed at T3) collected via phone interviews to solicit their opinion on the strengths, limitations and suggestions on how to improve the quality of the training and online resources. The specific questions asked to the clinicians related to the workshop and online resources are presented in S1 Appendix;

2. PTs' perception of the PDDM model's contribution to their clinical assessment procedures and to target adequate treatment assessed at T3 via phone interviews. The specific questions asked to the clinicians related to the PDDM model's contribution are presented in S2 Appendix;

**Model's preliminary effect on patients reported outcomes.** Our third objective was to explore the preliminary effect of the intervention. For each of their patients, the PTs had to provide data over a 6-week period (or less if the patient was discharged before the end of the 6-week period) through the online tool. Dyads (PTs and patients) had to complete the questionnaires and enter all the relevant information through the electronic tool. The documentation of the preliminary short-term (T2: initial visit; T3: +6 weeks following initial assessment) clinical effects of the model were assessed through the analysis of core outcome measures recommended by the literature for clinical trials on LBP [21, 22] and related to the specific elements contained within each of the five domains of the PDDM (Table 1).

**Other outcomes.** A strong association exists between poor sleep quality and increased pain and disabilities in the LBP population [29, 30]. There is also a high prevalence of depressive and anxiety symptoms/disorders among patients with chronic LBP [31]. Thus, we decided to capture these outcomes and opted to select specific items in the self-reported questionnaires already used rather than adding two additional validated measurement tools. Our pre-test involving a sample of PTs and patients informed us that a maximum of 20 minutes to complete all questionnaires was acceptable by both stakeholders. See Table 1 for detailed description of these outcomes (Table 1). It is important to note that the additional items used do not represent validated measures of these constructs, but rather serve as useful clinical tools to provide insight of the patient own perception about his sleep hygiene and mood.

Table 1. Preliminary effect outcome measures.

| Domain | Specific items | Outcome measures (T2 and T3) |
|---|---|---|
| Nociceptive pain drivers | Pain and impact of pain on physical function | *Brief Pain Inventory* (BPI) is designed to assess pain severity (at its worst, least and average) and the extent to which pain interferes in the daily life in relation to 7 domains of functioning (general activity, mood, walking ability, normal work, relations, sleep and enjoyment of life) on a scale of 0 to 10 [23]. |
| Nervous system dysfunction (NSD) drivers | Radicular signs/symptoms, hyperalgesia/allodynia, evidence of central sensitization | The *PainDetect Questionnaire* [24] is a reliable screening tool to predict the likelihood of a neuropathic pain component. The total score indicates if the pain is less likely to be neuropathic (i.e., 0–12 nociceptive pain; 13–18 mixed pain; 19–38 most likely neuropathic pain) |
|  |  | The *Central sensitization Inventory* (CSI) consists of a self-reported tool to assess symptoms of central sensitization (CS) [25, 26]. It contains two sections, part A and B. Part A contains 25 items with 5-point Likert scale with a range for the total score from 0 to 100 and is intended to give an overview of the symptoms that are common in CS. Part B was not used as an outcome measure since it only identifies if the patient has been diagnosed with specific disorders associated with CS. |
| Cognitive-emotional drivers | False beliefs, fear of pain/movement, self-efficacy, mood | The *STarT Back Tool* (SBT) is a screening questionnaire consisting of 9 items based on psychosocial factors used to categorize patients with LBP based on risk (low, medium, or high) for poor disability outcomes [27]. Overall scores (ranging from 0 to 9) are determined by summing all responses, and the SBT psychosocial subscale (items 5–9; ranging from 0 to 5) are determined by summing all items related to psychosocial factors of prolonged disability such as catastrophizing and pain-related fear and anxiety [27]. |
| Contextual drivers | Job flexibility, absenteeism, work capacity | The *Örebro Musculoskeletal Pain Screening Questionnaire Short-form* (OMPSQ-SF) is a 10-item screening questionnaire used to determine the risk of long-term absenteeism from work due to LBP based on occupational and social factors [28]. |
| Other outcomes | Sleep disturbances | "Sleep disturbances" were deemed "present" if the patient answered "often or always" to **item 1 of the CSI** ("I feel tired and unrefreshed when I wake from sleeping") and/or score $\geq 4$ on **item 13 of the BPI** ("during the past 24 hours, pain has interfered with your sleep"). |
|  | Mood, depressive symptoms | "Depressive symptoms" were deemed "present" if the patient answered "sometimes, often or always" to **item 16 of the CSI** ("I feel sad or depressed") and/or score $>5$ on **item 6 of the OMSPQ-SF** ("how much have been bothered by feeling depressed in the past week?") |

## Sample size

As this is a feasibility trial, no formal sample size calculation was conducted [32]. Instead, we aimed to recruit 10–15 clinicians in order to effectively measure the feasibility of the intervention and asked each therapist to provide complete data for at least three of their patients with LBP for an expected total of 30–45 patients.

## Data analysis

Participants' baseline characteristics are presented as the mean and standard deviation (SD) for continuous variables and frequency and percentage for categorical variables. The primary analyses for this trial focused on a description of the applicability of the model, feasibility of recruitment procedures, retention rates of participants, suitability of the eligibility criteria and compliance to the study protocol, and acceptability of the intervention based on the detailed description for each outcome in the Methods section. Clinicians' perceived acceptability of the use of the PDDM model in clinical practice was analyzed by qualitative reporting of data. Thematic analysis was conducted on the audio recordings of the semi-structured interviews in order to identify the predominant themes. Coding was performed manually by a member of the research team (FN) using a predefined coding list pertaining to the components of the semi-structured interviews (i.e., clinicians' appreciation of the training workshop, acceptability of the assessment procedures and the model's coherence and perceived effectiveness to help refine diagnostic and guide treatment). Emergent codes were sought in an effort to remain as faithful as possible to the opinions

expressed by the participants. A narrative summary based on the thematic analyses of the semi-structured individual interviews was written and presented in this study. This summary guided the decision to determine the acceptability of the PDDM model (or lack of) following a consensus among the members of the research team. Exploratory quantitative analyses were conducted to analyze the preliminary effect of the intervention on the selected patient clinical outcomes at T2 and T3. As this was a feasibility trial the objective was not hypothesis testing, rather these analyses allowed for the exploration of the preliminary effect of the PDDM model. Thus, we limited the analyses to pre-post intervention comparisons using paired t tests.

## Results

### Clinicians' characteristics

Twenty-four clinicians participated to the workshop and provided baseline data (Table 2). The majority of clinicians were women (15/24, 63%) and mean age was 35 years (SD = 9.7) with an average of 11 years of work experience (SD = 8.9). Fifteen clinicians (63%) worked in the public sector and most of them (21/24, 88%) already used classification systems in their clinical practice for the management of their LBP patients (e.g., mechanical diagnosis and therapy).

### Patients' characteristics

We obtained data from 61 patients. Thirty patients (49%) were women and the mean age of our sample was 51 years (SD = 15.0). The majority of our patients' sample had LBP for more than three months (n = 48, 79%). Detailed patients' baseline sociodemographic and clinical characteristics are presented in Table 3. The study flowchart is presented in Fig 1.

### Applicability of the PDDM model

The applicability of the PDDM model to a sample of patients with a primary complaint of LBP was confirmed. The model served to establish the profile of a patient according to the presence or absence of elements of each categories (A, B, A+B or none) for each domain. Indeed, each category of the 5 domains of the PDDM model was represented and evenly distributed in the patients' sample, providing evidence for the relevance of establishing the profile of patients according to the presence of more common and/or modifiable factors (A) and/or more complex elements (B) (Fig 2). For the nociceptive drivers domain, 60% of patients were responders

**Table 2. Characteristics of clinicians.**

| Categories | Clinicians (n = 24) |
|---|---|
| **Gender:** | |
| •Male | 9 |
| •Female | 15 |
| **Age (mean, SD)** | 35 (9.7) |
| **Years of work experience (mean, SD)** | 11 (8.9) |
| **Clinical setting:** | |
| •Public | 15 |
| •Private | 9 |
| **Education level:** | |
| •Bachelor | 13 |
| •Master | 11 |
| **Use of classification systems for LBP:** | |
| •Yes | 21 |
| •No | 3 |

**Table 3. Patients' sociodemographic and clinical characteristics at baseline.**

| Categories | Patients (n = 61) |
|---|---|
| **Gender:** | |
| • Male | 31 |
| • Female | 30 |
| **Age (mean, SD)** | 51 (15.0) |
| **Duration of back pain (n; % of sample):** | |
| • > 1 year | 28 (45.9%) |
| • 6–12 months | 5 (8.2%) |
| • 3–6 months | 15 (24.6%) |
| • 4–11 weeks | 5 (8.2%) |
| • 0–4 weeks | 8 (13.1%) |
| **BPI[a] (mean, SD):** | |
| • Pain severity | 4.6 (2.2) |
| • Pain interference | 4.6 (2.5) |
| **CSI[b] —total score (mean, SD):** | 39.1 (15.2) |
| • Subclinical | 24% |
| • Low | 27% |
| • Moderate | 21% |
| • Severe | 17% |
| • Extreme | 11% |
| **PainDetect questionnaire—total score (mean, SD):** | 13.4 (6.9) |
| • Nociceptive | 56.5% |
| • Neuropathic | 25.8% |
| • Unsure | 17.7% |
| **SBT[c] —total score (mean, SD):** | 3.0 (1.5) |
| • Low risk | 33.3% |
| • Medium risk | 27.4% |
| • High risk | 38.7% |
| **OMSPQ-SF[d] —total score (mean, SD):** | 50.9 (15.4) |
| • Below cut-off score (low risk) | 50.0% |
| • Above cut-off score (high risk) | 50.0% |
| **Sleep disturbances:** | |
| • Nothing to report ('no' to both items) | 19.4% |
| • "Often or always" to item 1 of the CSI | 17.7% |
| • Score $\geq$4 on item 13 of the BPI | 22.6% |
| • "Often or always" to item 1 of the CSI **AND** Score $\geq$4 on item 13 of the BPI | 40.3% |
| **Mood and/or depressive symptoms:** | |
| • Nothing to report ('no' to both items) | 41.9% |
| • "Sometimes, often or always" to item 16 of the CSI **OR** Score >5 on item 6 of the OMSPQ-SF | 38.7% |
| • "Sometimes, often or always" to item 16 of the CSI **AND** Score >5 on item 6 of the OMSPQ-SF | 19.4% |

[a]BPI = brief pain inventory

[b]CSI = central sensitization index

[c]SBT = start back stool

[d]OMSPQ-SF = örebro musculoskeletal pain questionnaire

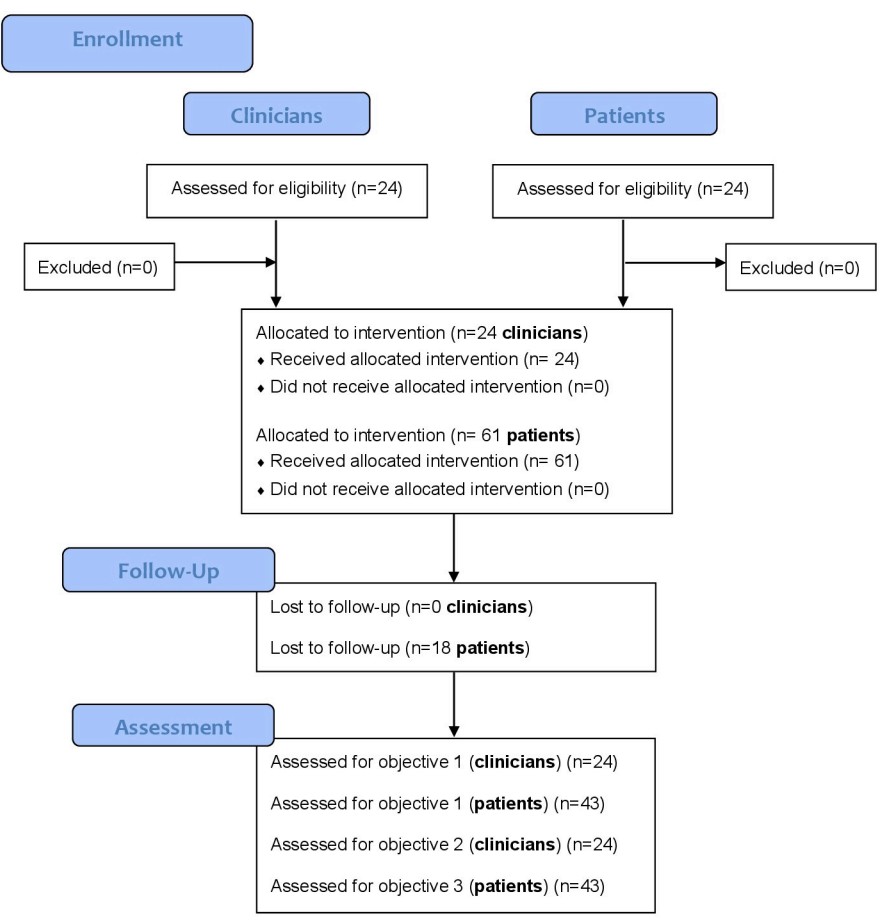

**Fig 1. Study flowchart.**

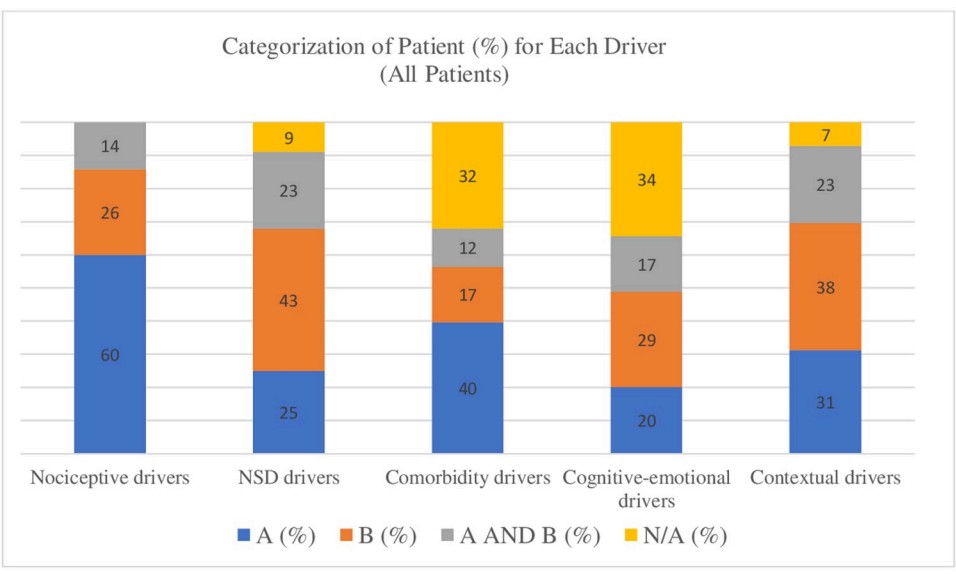

**Fig 2. Percentage of patients categorized in the different categories of each domain of the PDDM.**

to a classification system and classified in category A. The vast majority (91%) of the sample presented with the presence of elements from the nervous system dysfunctions domain. We noted that 43% of patients showed evidence of nervous system hypersensitivity. As for the comorbidity drivers domain, only 32% of the patients did not have any comorbidities, while 40% presented with physical comorbidities and 12% with both physical and mental-health comorbidities. We also observed that half of our sample (49%) presented with either maladaptive cognitive-emotional factors or malaptative pain behaviors, while a significative proportion of patients (34%) did not demonstrate evidence of cognitive-emotional drivers of pain and disability as tested with the electronic data collection tool based on the PDDM principles. As for the contextual factors domain, 91% of patients presented with elements supporting that contextual factors influenced their pain and/or disability, with 31% of patient dealing with work-related challenges and 23% presenting with drivers from both social and occupational contexts. In summary, the totals within each domain support the model's ability to capture each patient's profile according to the PDDM.

## Feasibility outcomes

Feasibility of the recruitment procedures was successful as both PTs' and patients' recruitment exceeded our preliminary expectations. Twenty-four (24) clinicians were recruited, of which nine came from the *private practice* network of clinics and 16 worked in various outpatient clinics of the *CIUSSSE (public network)*. All clinicians that expressed interest to the clinical managers of the participating clinics met the eligibility criteria and were enrolled in the study. However, the administrators did not report the number of clinicians who did not wish to participate in the study among all eligible clinicians working in their clinics. For patients, reasons for refusing to participate were only collected via the clinicians' interviews at the end of the study and included the burden (i.e., patient's condition too irritable) and additional time to answer all the questions and being incomfortable with technology. However, the number of patients who refused to participate was not collected by the clinicians, which represent a feasibility issue to consider for future trial. The clinicians recruited a total of 61 patients for an average of 2.5 patients per clinician (SD = 2.0). The recruitment ratio of patients per clinician was higher for the outpatient public clinics (3.2, SD = 2.0) compared to the private clinics (1.5, SD = 1.4). For the patients, an attrition rate of 29% was observed at 6-week follow-up (43/61 of patients' data were reported at follow-up). Unfortunately, reasons for and patterns of attrition were not collected due to issues encountered during the data collection by the clinicians. Only two clinicians (8%) did not report any patients' data after the workshop. These clinicians mentioned having forgotten the ongoing research project and administrative issues related to their employment status as reasons for not having recruited patients. The mean compliance (%) to the study protocol among all clinicians was 82% with a superior compliance observed in the public settings (92%) compared to the private clinics (67%). A little more than half of the clinicians (54%, 13/24) reported all outcome measures relevant to the study. Feasibility outcomes are detailed in Table 4. Eligibility criteria were deemed adequate since overall recruitment was successful for both PTs and patients and no issues were mentioned by the clinicians concerning the eligibility criteria during the interviews. However, a small minority of clinicians reported that they were more likely to select patients that appeared familiar with the use of technology.

## Acceptability outcomes

Thematic analysis of the interviews lead to the identification of three main themes:

**Table 4. Detailed feasibility outcomes for each participating clinical setting and clinicians.**

| Clinical setting | Clinicians | Outcomes | | | | Compliance to study protocol (%) | # of patients recruited |
|---|---|---|---|---|---|---|---|
| | | Interview T1 | Patient outcomes T2 | Interview T3 | Patient outcomes T3 | | |
| **Private practice network** | 1 | ✓ | ✓ | ✓ | ✓ | 100 | 3 |
| | 2 | ✓ | - | - | - | 25 | 0 |
| | 3 | ✓ | - | ✓ | - | 50 | 0 |
| | 4 | ✓ | - | - | - | 25 | 0 |
| | 5 | ✓ | ✓ | ✓ | - | 75 | 2 |
| | 6 | ✓ | ✓ | ✓ | ✓ | 100 | 3 |
| | 7 | ✓ | ✓ | ✓ | - | 75 | 2 |
| | 8 | ✓ | - | ✓ | - | 50 | 0 |
| | 9 | ✓ | ✓ | ✓ | ✓ | 100 | 3 |
| **Public network** | 10 | ✓ | ✓ | ✓ | ✓ | 100 | 4 |
| | 11 | - | ✓ | ✓ | ✓ | 75 | 3 |
| | 12 | ✓ | ✓ | ✓ | ✓ | 100 | 3 |
| | 13 | ✓ | ✓ | ✓ | ✓ | 100 | 4 |
| | 14 | ✓ | ✓ | ✓ | ✓ | 100 | 3 |
| | 15 | ✓ | ✓ | ✓ | ✓ | 100 | 3 |
| | 16 | ✓ | ✓ | ✓ | ✓ | 100 | 3 |
| | 17 | ✓ | ✓ | ✓ | ✓ | 100 | 5 |
| | 18 | ✓ | ✓ | - | ✓ | 75 | 4 |
| | 19 | ✓ | - | ✓ | ✓ | 75 | 1 |
| | 20 | ✓ | ✓ | ✓ | ✓ | 100 | 1 |
| | 21 | ✓ | ✓ | ✓ | - | 75 | 2 |
| | 22 | ✓ | ✓ | ✓ | ✓ | 100 | 1 |
| | 23 | ✓ | ✓ | ✓ | ✓ | 100 | 9 |
| | 24 | ✓ | ✓ | ✓ | - | 75 | 2 |
| | Total | 23/24, 96% | 19/24, 79% | 21/24, 88% | 17/24, 71% | 82% | Total = 61* |

✓ = participant reported the selected outcome

- = participant did not reported the selected outcome

* = Mean number of patient/clinician = 2.5

**Clinicians' appreciation of the workshop.** The one-day workshop format was well appreciated by all the clinicians. They mentioned that the order in which the different aspects were presented made sense to them since they were presented with the model's theoretical foundations first, and then were able to apply their learnings to practical case studies. The duration and "intensity" of the workshop was acceptable, and all participants agreed that a one-day format is optimal compared to half a day (too short) or more than one day (too long).

Several strengths of the training were highlighted by the participants and include the use of relevant case studies that facilitated the clinical application of the model, the presentation of various relevant self-reported questionnaires to use in practice, the evidence-based nature of the information given and the interactive training structure, which made the participants feel comfortable to ask questions. Participants also expressed some limitations and suggested recommendations (Table 5). They raised the importance of putting more emphasis on giving tangible examples of intervention strategies related to the application of the model and providing more instructions on the interpretation of questionnaires presented during the workshop. Participants also suggested the idea of providing preparatory documentation to reduce the

Table 5. Limitations and recommendations reported by the clinicians.

| Limitations raised by the clinicians | Potential solutions and recommendations made by clinicians |
|---|---|
| • Interpretation of the questionnaires' scores | • Provide more guidance on the use and interpretation of questionnaires' scores during workshop |
| • Not enough emphasis on intervention strategies, especially for psychosocial factors | • Develop additional ressources on intervention strategies for psychosocial factors (i.e., short video capsules on how to deal with psychosocial factors suchs as pain catastrophizing, self-efficacy, gradual exposure, etc.) |
| • More time for the clinical integration of the PDDM model would be ideal | • Provide preparatory documentation prior to workshop |
| • Means of delivery (i.e., unreliable internet access, time to complete questionnaires for vulnerable/specific patients) | • Develop an offline version of the electronic tool<br>• Reduce the amount of items<br>• Provide the questionnaires prior to patient appointment |

amount of information given during the workshop and to allow more time for clinical application of the model through additional case studies. All clinicians mentioned that the online supplementary resources (i.e., website, mind map, links to validated questionnaires) were relevant, user-friendly, and that they had consulted them throughout the research project to facilitate the model's clinical application.

**Clinicians' perception of the PDDM model's contribution to their clinical practice.** All clinicians deemed the model relevant to perform a more comprehensive assessment of their patient. A common theme among the participants was the PDDM model's contribution to the assessment of psychosocial and prognostic factors, where the model helped them guide their assessment procedures by its ability to objectify psychosocial factors that were initially more of a subjective impression to the participants. Indeed, they mentioned that the model helped them improve their lack of understanding related to the identification of psychosocial factors, thus favoring a more complete clinical profile, which in turn helped them deliver a more personalized treatment plan. The model was also considered appropriate by the clinicians to target adequate treatment procedures. The additional information gathered via the PDDM model enabled them to enhance and personalize their education intervention and exercise programs based on the patient's needs in addition to being more attentive to the necessity of adopting a more psychologically informed approach for patients with a high risk of chronicization. According to the participants, the PDDM model facilitated the delivery of personalized care, referral to another professional when necessary, interprofessional communication and patient education. However, clinicians reported a lack of resources to adequately manage psychosocial factors on their own after effectively identifying them (Table 5).

**Acceptability and suitability of the online tool hosted on LimeSurvey®.** The online tool, hosted on LimeSurvey® to support the integration of the PDDM model into the practice of the clinicians, was considered easy to use and an appropriate tool to facilitate the model's integration into practice. The time needed for its completion by the clinician was judged adequate, but the time and the means of delivery could represent a barrier for patients with pain-related attention deficits or those that are not familiar with technology, as they might be more refractory to its use (Table 5). According to the clinicians, the additional time needed to complete the survey had no significant effect on their clinical encounter since it helped them save precious time and efforts on the assessment of more time-consuming aspects such as psychosocial and prognostic factors. However, challenges to the integration of the model in practice were noted by the majority of the clinicians. First, unreliable internet access represented a

barrier for a number of public outpatient clinics in the *CIUSSSE*, which complicated the use of the electronic tool (Table 4). Second, in order to have the same amount of time with their patients during the initial visit, some clinicians asked their patients to arrive a few minutes early to their appointment to complete the questionnaires, which was not possible for all patients (Table 5). Some clinicians suggested during the interviews (T3) to send all self-reported questionnaires via email prior to their initial appointment so that the patient could complete them at home (Table 5).

**Preliminary effect outcomes.** A positive effect was observed for all patients' reported outcome measures. Detailed results for each outcome measure at baseline and at follow-up are presented in Table 6. All outcome measures, except for the PainDetect questionnaire showed a

**Table 6. Preliminary effect outcome measures at baseline and follow-up.**

| | Outcomes measures | Baseline mean (SD) n = 43 | 6-week follow-up mean (SD) n = 43 | Mean of individual differences (T3-T2) (mean ± 95% CI) |
|---|---|---|---|---|
| BPI[a] | Pain severity | 4.6 (2.4) | 3.1 (2.3) | -1.53 [-1.95, -1.11] p<0.001 |
| | Pain interference | 4.6 (2.6) | 3.2 (2.7) | -1.39 [-1.86, -0.91] p<0.001 |
| CSI[b] | Total score | 38.7 (14.2) | 35.0 (15.9) | -3.72 [-6.31, -1.14] p = 0.006 |
| | **Subclinical** | **24%** | **40%** | **+16.2%** |
| | Low | 27% | 20% | -7% |
| | Moderate | 21% | 24% | 3.8% |
| | **Severe** | **17%** | **9%** | **-8.6%** |
| | **Extreme** | **11%** | **7%** | **-4.4%** |
| PainDetect questionnaire | Total score | 13.8 (7.3) | 12.0 (7.7) | -1.76 [-3.55, 0.03] p = 0.054 |
| | Nociceptive | 56.5% | 54.5% | 2.2% |
| | Neuropathic | 25.8% | 18.2% | -7% |
| | Unsure | 17.7% | 25.0% | 7.3% |
| SBT[c] | Total score | 3.0 (1.5) | 2.0 (1.7) | -0.91 [-1.3, -0.49] p<0.001 |
| | **Low risk** | **33.3%** | **47.7%** | **+14.4%** |
| | Medium risk | 27.4% | 29.5% | +2.1% |
| | **High risk** | **38.7%** | **22.7%** | **-16.0%** |
| OMSPQ-SF[d] | Total score | 52.0 (16.1) | 42.4 (18.9) | -9.53 [-13.17, -5.90] p < 0.001 |
| | **Below cut-off score (low risk)** | **50.0%** | **62.0%** | **+12.0%** |
| | Above cut-off score (high risk) | 50.0% | 38.0% | -12.0% |
| Sleep disturbances | Nothing to report ('no' to both items) | 19.4% | 43.2% | +23.8% |
| | "Often or always" to item 1 of the CSI ("I feel tired and unrefreshed when I wake from sleeping") | 17.7% | 11.4% | -6.3% |
| | Score ≥4 on item 13 of the BPI ("during the past 24 hours, pain has interfered with your sleep") | 22.6% | 15.9% | -6.7% |
| | "Often or always" to item 1 of the CSI **AND** Score ≥4 on item 13 of the BPI | 40.3% | 29.5% | -10.8% |
| Mood and/or depressive symptoms | Nothing to report ('no' to both items) | 41.9% | 56.8% | +14.9% |
| | "Sometimes, often or always" to item 16 of the CSI ("I feel sad or depressed") **OR** Score >5 on item 6 of the OMSPQ-SF ("how much have you been bothered by feeling depressed in the past week?") | 38.7% | 29.5% | -9.2% |
| | "Sometimes, often or always" to item 16 of the CSI ("I feel sad or depressed") **AND** Score >5 on item 6 of the OMSPQ-SF ("how much have you been bothered by feeling depressed in the past week?") | 19.4% | 13.6% | -5.8% |

[a]BPI = brief pain inventory

[b]CSI = central sensitization index

[c]SBT = start back stool

[d]OMSPQ-SF = örebro musculoskeletal pain questionnaire

statistically significant reduction post-intervention (T3) (p<0.05) (Table 6). For the self-reporting of sleep disturbances and depressive symptoms, an increase of 23% and 15% of patients having nothing to report were observed respectively pre-to-post intervention (Table 6).

## Discussion

This is the first feasibility trial of our recently proposed rehabilitation management model for LBP—the PDDM model. Our aims were to determine the applicability of the model to a LBP population, to document the feasibility of conducting a pragmatic trial and to explore clinicians' perceived acceptability of the PDDM model's use in clinical settings and its short-term effect on patient outcomes. This led to several observations. Overall, the applicability of the PDDM model was confirmed, the trial design was found to be feasible and the use of the PDDM to guide the clinical assessment procedures and to target adequate treatment have been deemed acceptable by rehabilitation professionals working with patients experiencing LBP contingent upon some modifications to the study design. We met our predefined success criteria for the feasibility outcomes and recommendations to enhance data collection methods were highlighted. The target number of clinicians and patients were recruited within the trial timeframe, the suitability of the eligibility criteria was confirmed and the attrition rate of participants and the compliance rate of clinicians to the study protocol were acceptable, although just below the pre-defined threshold value for success criterion for attrition. Clinicians' acceptability of the use of the model in clinical settings and their appreciation of the training were both positive. Recommendations to improve the model's integration in clinical practice and content of the workshop were identified, which will prove useful for a future larger trial.

Preliminary effects of the model have shown positive and promising results on various relevant LBP-related outcomes. However, caution is required when interpreting the exploratory analysis of the model's clinical outcomes as, due to the feasibility aims and nature of this trial, it was not designed nor intended to determine effectiveness. Moreover, the relatively high attrition rate observed would tend to undermine the findings. Our preliminary findings precede a larger scale trial that may likely impact clinical practice based on the promising results of this study. Additional research on the model's contribution to the clinical decision-making process of rehabilitation professionals and its effectiveness are needed to warrant such claims.

### Discrepancies between clinical settings

An interesting finding was the difference between the private and public clinical settings in terms of patients recruited per clinician ratio and clinicians' compliance to the study protocol in favor of the public settings. A possible explanation for this discrepancy includes resource constraints in part due to busy schedules, limited within-session time and a possible lack of commitment and/or interest from clinicians working within the private sector for a model more adapted for patients presenting with a more complex profile who might not represent a major part of their clientele. Administrative particularities such as pay structure (i.e., public employees paid on salary while doing study tasks unlike clinicians working in the private sector who might be paid per patient) could help explain this finding. However, the final interviews with the participants did not reveal any specific causes to help explain these discrepancies between settings.

### Relevance of the PDDM model in determining patient's phenotypes

The findings of this study demonstrate the ability of the PDDM model to adequately measure the relative contribution of each domain to create a more complete clinical profile or

phenotype with a sample of patients with a primary complaint of LBP. Using the phenotype of the patient to guide assessment procedures, treatment, prognostic and patient outcomes have been gaining popularity in the MSK care literature owing to the high heterogeneity of patients presenting with MSK-related disorders and clinicians' reliance on oversimplified MSK diagnoses that limit their capacity to help their patients [33]. This concept has been recommended by international experts in the field of pain research [34]. Diagnostic frameworks such as the PDDM model studied in this paper represent an interesting avenue [33].

## Contribution of the PDDM model to clinicians' practice

A common theme that emerged from the phone interviews with the clinicians at T3 was the significative contribution of the model in assessing and objectifying psychosocial factors that are present in patients with LBP. They did not feel as confident in assessing these factors as compared to the mechanical aspects of LBP, a common preference for PTs [35]. These observations are also present in the % of patients categorized in the different categories for domain 4 and 5 (cognitive-affective and contextual drivers). A vast majority of the patients in the study sample were classified as having at least one or more psychosocial factors contributing to the patient pain and disability, which highlights the importance of the issues reported by PTs with dealing with these factors. Moreover, the cognitive-affective drivers domain reported the highest % of patients presenting no elements of either category, which could be partly explained by the difficulties encountered by PTs to assess psychosocial factors. These difficulties in the assessment of psychosocial status by rehabilitation professionals have also been reported in the literature—a qualitative study reported a poor understanding of the role of psychosocial factors in the patient's clinical presentation and a lack of knowledge about their assessment [36]. A systematic review and qualitative meta-analysis exploring PTs' perceptions about the cognitive, psychological and social factors that may act as barriers to recovery for people with LBP reported that they felt unprepared to assess these factors and tended to stigmatize patients who reported behaviors suggestive of them [35]. A recent shift towards a more biopsychosocial approach by PTs has been observed but the training interventions seem to be insufficient to help them feel confident in delivering all the aspects associated with this perspective [37]. Based on the preliminary findings of this study, the PDDM model represents an interesting solution to this issue. The biopsychosocial lens embedded in the ICF, which also serves as the theoretical underpinning of the PDDM model represent a major strength of this newly developed management model, which could ultimately improve the rehabilitation management of patients living with LBP and presenting with psychosocial obstacles to recovery.

## Strengths and limitations

This trial has a number of strengths. This is the first time exploring the applicability, feasibility, acceptability and preliminary effect of the recently validated PDDM model. The pragmatic design of the trial was instrumental in meeting the success feasibility criteria since it considered the challenges associated with the reality of standard clinical practice such as busy schedules and the administrative particularities of each clinical setting. Clinical settings from both the public and private sectors were recruited, which shed lights on potential challenges for future studies inherent to these two distinct environments. However, this study also has limitations. The study design did not include a control or comparator group therefore the observed changes pre- and post-intervention may not be attributed solely to the intervention. We must keep in mind that natural history or a maturation effect could partly explain the positive changes [38]. However, the vast majority of our sample has reported having pain for a longer time with 77% (48/62) of patients reporting pain for more than 3 months. That being said, an

additional measurement time (i.e., 12 weeks or even longer) and the introduction of a control group (i.e., a two-arm controlled trial) could represent an interesting avenue for a future trial in order to take into consideration the good initial evolution associated with LBP and to comprehensively assess the model's effectiveness pre-post intervention. The convenience sampling method used to recruit both the clinics and clinicians could have introduced a potential selection bias. However, this is mitigated by the inclusion of various clinical settings from both the private and public settings and different demographic areas. No formal knowledge assessment was conducted following PTs' training on the PDDM model, which could optimize adequate comprehension and use of the PDDM model.

We identified some limitations to be addressed in a future trial, which specifically concern the data collection method as this aspect was deemed more problematic based on the study findings. Clinical administrators and clinicians did not collect data on the eligible PTs and patients that refused to participate in the study, thus making it impossible to report data on the recruitment rates of participants. Only general explanations from clinicians during the T3 interviews shed some lights on patients' refusals to participate. We initially planned to collect patients' sociodemographic characteristics (i.e., education, marital status, occupation, etc.) through the health questionnaires already used in the clinical settings by clinicians at baseline to be consistent with a pragmatic trial, which revealed not to be feasible in the context of this study. This led to an incomplete description of our sample. Additionally, clinicians were responsible to collect patients' outcomes measures at six weeks. This has also proven to be problematic since patients' losses to follow up due to not being reachable by the clinicians and/or simple oversights were reported. Other possible explanations are patients being referred elsewhere or patients not showing up to appointments, but were not documented. These limitations led to the main issue of not being able to differentiate between patients' drop-outs and losses to follow up within the 18 patients that did not report data at T3. Moreover, reasons for and patterns of attrition were not documented by the clinicians, thus differences between completers and non-completers were not assessed by the clinicians. This is particularly important considering the level attrition reported in this study. However, these issues could be easily resolved based upon few modifications to the data collection process and study design since they may be explained by the increased burden and reliance on the clinicians to collect all this data.

## Recommendations for a future larger scale trial

Based on the limitations discussed above and the feasibility and acceptability findings, a list of recommendations for a future larger scale trial has been documented:

- Clinical administrators and clinicians should be alerted to the importance of collecting the information on eligible participants that refuse and/or accept to participate in the study to inform the recruitment rate and potential refusals;

- The research team should develop a comprehensive and detailed plan to collect data directly from the patients themselves by taking charge of the follow-ups by means of reminder emails and reaching out directly to the patients to limit and/or document losses to follow-ups or potential drops out. This would also alleviate the clinicians' workload related to their participation in the study and facilitate future recruitment.

- Comprehensive reporting of patients' sociodemographic characteristics at baseline by the research team is essential to provide a clear description of the study sample. Collecting patients' coordinates (e.g., email addresses) to establish a direct link with them at follow-ups should also be encouraged.

- A formal knowledge and skills assessment following the workshops should be conducted in order to optimize adequate understanding and use of the PDDM model in clinical practice, thus fostering fidelity of intervention.

- In order to determine the effectiveness of the PDDM model, a future larger scale trial with a control group should be conducted. Randomization should also be considered to minimize selection bias.

## Conclusion

This study helped determine important factors imperative in the consideration of a future larger scale pragmatic trial design such as the PDDM model's applicability to a LBP population, clinicians' acceptability of the model, feasibility of the trial design and the model's preliminary effect on various relevant clinical outcome measures associated with LBP. Findings support the applicability and acceptability of the integration of the model by the clinicians and the feasibility of conducting such trial contingent upon a few modifications. It provides preliminary evidence of the high potential of the PDDM model to optimize LBP management as well as clear recommendations for conducting a future main study to explore its effectiveness. By assessing the applicability, feasibility, acceptance, and preliminary effects of the intervention for this population, we were able to gain a key insight as to how such an intervention can be best delivered to PTs in order to foster its implementation in clinical settings.

## Supporting information

**S1 Table. CONSORT 2010 checklist for randomized pilot and feasibility trials.**
(PDF)

**S2 Table. TREND checklist.**
(PDF)

**S3 Table. Detailed description of the categories for each domain of the PDDM model.**
(DOCX)

**S1 Fig. Pain and disability drivers management model.**
(TIF)

**S2 Fig. Example of a patient's clinical profile based on the PDDM model.**
(TIF)

**S3 Fig. Study timeline.**
(TIF)

**S1 Appendix. Questions asked to clinicians on their appreciation of the workshop (T1) and online resources (T3).**
(PDF)

**S2 Appendix. Questions asked to clinicians on the PDDM model's contribution to their practice.**
(PDF)

**S1 Protocol.**
(DOCX)

## Acknowledgments

We would like to thank the management and the participating physiotherapy professionals of the PhysioExtra Clinics (https://physioextra.ca/en/clinics/) and the CIUSSSE—Centre Hospitalier Universitaire de Sherbrooke (https://www.santeestrie.qc.ca/en/home/) for their support and participation in this research project.

## Author Contributions

**Conceptualization:** Christian Longtin, Simon Décary, Chad E. Cook, Marc O. Martel, Yannick Tousignant-Laflamme.

**Data curation:** Christian Longtin, Florian Naye, Yannick Tousignant-Laflamme.

**Formal analysis:** Christian Longtin, Florian Naye, Yannick Tousignant-Laflamme.

**Funding acquisition:** Yannick Tousignant-Laflamme.

**Investigation:** Christian Longtin, Simon Décary, Florian Naye, Yannick Tousignant-Laflamme.

**Methodology:** Christian Longtin, Simon Décary, Chad E. Cook, Florian Naye, Yannick Tousignant-Laflamme.

**Project administration:** Christian Longtin, Yannick Tousignant-Laflamme.

**Resources:** Christian Longtin, Yannick Tousignant-Laflamme.

**Software:** Christian Longtin, Yannick Tousignant-Laflamme.

**Supervision:** Christian Longtin, Simon Décary, Chad E. Cook, Yannick Tousignant-Laflamme.

**Validation:** Christian Longtin, Simon Décary, Chad E. Cook, Marc O. Martel, Sylvie Lafrenaye, Lisa C. Carlesso, Yannick Tousignant-Laflamme.

**Visualization:** Christian Longtin, Sylvie Lafrenaye, Lisa C. Carlesso, Yannick Tousignant-Laflamme.

**Writing – original draft:** Christian Longtin, Yannick Tousignant-Laflamme.

**Writing – review & editing:** Christian Longtin, Simon Décary, Chad E. Cook, Marc O. Martel, Sylvie Lafrenaye, Lisa C. Carlesso, Florian Naye, Yannick Tousignant-Laflamme.

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
