## [Decision Letter · Decision Letter 0]

7 Oct 2020

PONE-D-20-22177

Optimizing management of low back pain through the Pain and Disability Drivers Management Model: a feasibility trial

PLOS ONE

Dear Dr. Tousignant-Laflamme,

Thank you for submitting your manuscript to PLOS ONE. After careful consideration, we feel that it has merit but does not fully meet PLOS ONE’s publication criteria as it currently stands. Therefore, we invite you to submit a revised version of the manuscript that addresses the points raised during the review process.

We look forward to receiving your revised manuscript.

Kind regards,

Jose María Blasco, Ph.D.

Academic Editor

PLOS ONE

Journal Requirements:

2. Please address the following:

- Please ensure you have thoroughly discussed all potential limitations of this study within the Discussion section, including the potential introduction of bias during sampling.

- Please include additional information regarding the survey or questionnaire used in the study and ensure that you have provided sufficient details that others could replicate the analyses. For instance, if you developed a questionnaire as part of this study and it is not under a copyright more restrictive than CC-BY, please include a copy, in both the original language and English, as Supporting Information.

- In your Methods section, please provide additional information about the demographic details of your participants. Please ensure you have provided sufficient details to replicate the analyses such as a table of relevant demographic details.

3.Thank you for submitting your clinical trial to PLOS ONE and for providing the name of the registry and the registration number. The information in the registry entry suggests that your trial was registered after patient recruitment began. PLOS ONE strongly encourages authors to register all trials before recruiting the first participant in a study.

a) your reasons for your delay in registering this study (after enrolment of participants started);

b) confirmation that all related trials are registered by stating: “The authors confirm that all ongoing and related trials for this drug/intervention are registered”.

Please also ensure you report the date at which the ethics committee approved the study as well as the complete date range for patient recruitment and follow-up in the Methods section of your manuscript-

Reviewers' comments:

Reviewer's Responses to Questions

**Comments to the Author**

1. Is the manuscript technically sound, and do the data support the conclusions?

Reviewer #1: Partly

Reviewer #2: Partly

2. Has the statistical analysis been performed appropriately and rigorously? 

Reviewer #1: No

Reviewer #2: I Don't Know

3. Have the authors made all data underlying the findings in their manuscript fully available?

Reviewer #1: No

Reviewer #2: Yes

4. Is the manuscript presented in an intelligible fashion and written in standard English?

Reviewer #1: Yes

Reviewer #2: Yes

5. Review Comments to the Author

Reviewer #1: This paper reports on a small feasibility study of integrating a pain management approach to standard care.

The aims of the feasibility study are clearly stated - the issue is that outcomes c and d are a bit unclear. For c, how is success defined here - how is it measured? For d, it's unclear how the interaction between clinician compliance and patient dropout in b is assessed - if a clinician loses their patient because of dropout what happens? And is the compliance rate by clinician or by patient?

Similarly in terms of acceptability what was the threshold for the whole thing being acceptable?

There is a clear issue here that needs to be addressed in terms of whether dropout here is associated with outcome - was data collected to see the pain score before dropout? With a third (nearly) of patients dropping out, if this is related to outcome or indeed baseline levels then this data is crucial. Table 3 for example needs to have the baseline values and sd for the ones with analysdable data - and an investigation of any differences between completers and non-completers. Given the 5 and 20 rule for missing data, the amount of missing data here is too large for reliable inference; while this is not required here, reasons for and patterns of attrition are key in a feasibility trial as this amount of missing data would tend to undermine the findings of a larger trial. Was retrieved dropout attempted and if not why not?

Please give exact p-values. Star notation has been frowned upon since the 1990s.

Given that a sixth of practices failed to recruit what were the barriers to recruitment? How many patients were seen and refused to participate?

Reviewer #2: This paper has seemed very interesting to me since it proposes the use of a classification and intervention system that is so necessary in patients with LBP. However, please consider the following comments about the study:

1. Line 76: Please consider to add "based exclusively on biological aspects" after "the use of classification systems..."

2. Even in a feasibility trial, please consider doing a sample size calculation/estimation.

3. In the "participants" section, please consider adding criteria related to the characteristics of the LBP (i.e., duration of symptoms [acute, sub-acute or chronic LBP specifying a measurement time] or specific vs non-specific LBP).

4. "Study setting" section is actually a description of the recruitment. Please consider transferring this section to the "Recruitment" section.

5. Line 141: "Clinic managers informed the clinicians working at the different sites." Please explain what the clinic managers informed the clinicians about.

6. Is a 1-day workshop enough to provide the clinicians the knowledge and skills they need to achieve the 4 objectives? In my opinion, this training should be carried out in greater detail. Also, authors should consider assessing the competence of clinicians managing the PDDM model after the training process in order to ensure the correct understanding and use of the tool.

7. Both applicability of the PDDM model and the feasibility of participants recruitment are not rigorously described in the "Data analysis" section. How the authors have analyzed those outcome measures?

8. There is a lack of information about the operation of the model. Please consider describing how the PDDM model classifies patients based on the five domains and what intervention strategies the PDDM suggests based on each category of the model.

9. Please review Table 5. It is written "psyschosocial facors" instead of "psychological factors".

10. Please consider including a control group in order to be able to correctly interpret a pre-post analysis.

6. PLOS authors have the option to publish the peer review history of their article (what does this mean?). If published, this will include your full peer review and any attached files.

Reviewer #1: No

Reviewer #2: No

---

## [Author Response · Author response to Decision Letter 0]

18 Nov 2020

Rebuttal letter – responses to reviewers 

We would like to thank the reviewers for their insightful comments. We have provided a detailed response (in red) to each comment/concern made by the editor and reviewers. 

Points raised by the academic editor and journal requirements: 

Please ensure that your manuscript meets PLOS ONE's style requirements, including those for file naming. The PLOS ONE style templates can be found at (links to PDF files). 

Response: We performed a thorough review of the style requirements and file naming according to the PLOS ONE style templates to meet the journal requirements. 

Please address the following: 

Please ensure you have thoroughly discussed all potential limitations of this study within the Discussion section, including the potential introduction of bias during sampling: 

Response: We thank the editor for this constructive comment. After a thorough revision process based on the reviewers’ comments, we have added considerable details concerning the potential limitations of this study within the Discussion section in the revised manuscript. More so, we provided additional information concerning the recommendations resulting from the study findings for a future trial. Thus, we now believe that we have thoroughly discussed all potential limitations in addition to providing clear solutions to ensure feasibility of a future trial. 

Please include additional information regarding the survey or questionnaire used in the study and ensure that you have provided sufficient details that others could replicate the analyses. For instance, if you developed a questionnaire as part of this study and it is not under a copyright more restrictive than CC-BY, please include a copy, in both the original language and English, as Supporting Information. 

Response: In order to provide additional details to ensure that others could replicate the analyses, we revised Supporting Information 6 (S6 Table) to clearly describe the process used to establish the patient’s profile according to the PDDM framework (intervention). We also added more precise and clear information concerning the method used to categorize patients in the revised manuscript in subsection Acceptability outcomes in the Methods section. 

In your Methods section, please provide additional information about the demographic details of your participants. Please ensure you have provided sufficient details to replicate the analyses such as a table of relevant demographic details. 

Response: Two additional tables were added (now labeled Table 2 and Table 3) in order to provide full details on participants’ (clinicians and patients) demographic and clinical characteristics at baseline. Please see the Results section of the revised manuscript. 

Thank you for submitting your clinical trial to PLOS ONE and for providing the name of the registry and the registration number. The information in the registry entry suggests that your trial was registered after patient recruitment began. PLOS ONE strongly encourages authors to register all trials before recruiting the first participant in a study. As per the journal’s editorial policy, please include in the Methods section of your paper: 

Your reasons for your delay in registering this study (after enrolment of participants started). 

Response: After thorough verification, we realized that we encountered some difficulties to register our trial on clinicaltrials.gov. Registration of the study was completed in March 2020 before patient recruitment began (emails as proof). However, when completing the submission process for this journal, we realized that we never received an official trial registration confirmation number. Thus, we went back to check on the status of the trial registration process on the website and completed the missing information which ultimately led to the trial registration number submitted with this study. This explains why the registry entry suggests that our trial was registered after patient recruitment began, which is not the case. A brief explanation was added to the revised manuscript in the Methods section under the subsection “Design”. 

Confirmation that all related trials are registered by stating: “The authors confirm that all ongoing and related trials for this drug/intervention are registered”. 

Response: The statement was added to the revised manuscript. See subsection “Design” of the “Methods” section of the manuscript. 

Please also ensure you report the date at which the ethics committee approved the study as well as the complete date range for patient recruitment and follow-up in the Methods section of your manuscript. 

Response: This information was added to the revised manuscript in the subsection “Recruitment” of the Methods section. 

Points raised by reviewer #1: 

The aims of the feasibility study are clearly stated - the issue is that outcomes c and d are a bit unclear. 

For c, how is success defined here - how is it measured? 

Response: We thank the reviewer for his comment as we realize that the success criteria for the feasibility outcomes were not clearly described in the original version of the manuscript. We provided more clarification on this issue in the revised manuscript. See the Feasibility outcomes section. See also the subsection Feasibility outcomes in the Results section for the reporting of these findings. 

For d, it's unclear how the interaction between clinician compliance and patient dropout in b is assessed - if a clinician loses their patient because of dropout what happens? And is the compliance rate by clinician or by patient? 

Response: Again, we thank the reviewer for his constructive comment. Unfortunately, we were not able to assess the interaction between clinician’s compliance and patient dropout. More precisely, it was not possible to state if the patient dropped out (loss during follow-up) of the study during the rehabilitation process. This is explained by the fact that the clinicians did not report this information and we did not collect the patients coordinates (personal contact info) to reach them directly since we originally thought that the clinicians would take note of this. Therefore, we could not differentiate dropouts from lost to follow-up, and decided to categorize all patients that did not report data at T3 as lost to follow-up. We acknowledge that this is an important limitation to this study, which is now further emphasized in the revised manuscript in the discussion. However, this preliminary feasibility study provides great insight on how we can improve the study design for future trials. For example, this limit could be addressed by the presence of a research assistant in charge of contacting each patient individually for follow-ups without having to put that burden on the clinician to collect all this information. This solution will be used in a future trial. To answer the reviewer’s second question, the compliance rate was calculated “by clinician”. This information was added to the revised manuscript. 

Similarly in terms of acceptability what was the threshold for the whole thing being acceptable? 

Response: No a priori threshold was set since acceptability was measured by means of qualitative methodology. Thematic analysis was performed on the verbatim of the semi-structure individual phone interviews. A narrative summary based on the thematic analysis of the transcript was written and presented in this study, which led us to believe that the intervention was acceptable. More details were added to the revised manuscript to better detail the acceptability assessment process. See subsection Data Analysis in the Methods section. 

There is a clear issue here that needs to be addressed in terms of whether dropout here is associated with outcome. 

Was data collected to see the pain score before dropout? With a third (nearly) of patients dropping out, if this is related to outcome or indeed baseline levels then this data is crucial. 

Response: As mentioned in our response to comment 1.b), we were not able to determine if the patient dropped out of the study or was lost to follow-up since the clinicians were not able to ensure efficient follow-ups with their patients and collect the relevant information (i.e., reasons for dropping out if that was the case). Therefore, it would be erroneous to assume that the patients actually dropped out of the study, hence why we did not report baseline scores specifically for these patients in our sample. Again, we acknowledge that is an important issue and we provided much more details in the discussion section. We agree with the reviewer that this information is essential and subsequent modifications to the protocol will be made in a future trial to collect this data. 

Table 3 for example needs to have the baseline values and sd for the ones with analysable data - and an investigation of any differences between completers and non-completers. 

Response: Table 3 was entirely revised based on the reviewer’s comments and a complete revision of the data by the research team. It now shows accurate findings for the patients with analysable data. An investigation of any differences between completers and non-completers was not carried out since it was not possible to clearly state if the non-completers dropped out of the study or were lost to follow-ups due to the inability of the clinicians to ensure adequate collection of this information as mentioned. 

Given the 5 and 20 rule for missing data, the amount of missing data here is too large for reliable inference; while this is not required here, reasons for and patterns of attrition are key in a feasibility trial as this amount of missing data would tend to undermine the findings of a larger trial. Was retrieved dropout attempted and if not why not? 

Response: We acknowledge that the amount of missing data due to attrition (or lost to follow-up) is quite high (roughly 30%). However, for patients with reported data at both T2 and T3, no missing data was reported due to the nature of the comprehensive online data collection tool. Therefore, missing data is entirely due to attrition in this study. As previously mentioned, we were not able to collect comprehensive data on reasons for and patterns of attrition since we did not collect patients coordinates to contact them directly as it was planned that clinicians would collect that data. However, additional information concerning feasibility outcomes were added to the revised manuscript based on the reviewers’ comments. It includes more detailed information concerning patients’ and clinicians’ reasons for refusing to participate in the study and feasibility issues with study design and data collection method. Finally, retrieved dropout was not attempted as the patients’ coordinates were not collected for this study, an aspect that will be addressed in a future trial. 

Please give exact p-values. Star notation has been frowned upon since the 1990s. 

Response: Exact p-values were added to Table 6 of the revised manuscript. 

Given that a sixth of practices failed to recruit what were the barriers to recruitment? How many patients were seen and refused to participate? 

Response: Additional information concerning patients’ refusal to participate and clinicians’ reasons for not recruiting patients were added in the Feasibility outcomes subsection in the Results section in the revised manuscript. 

Points raised by reviewer #2: 

Line 76: Please consider to add "based exclusively on biological aspects" after "the use of classification systems..." 

Response: We thank the reviewer for this suggestion. We believe it makes a good addition to the manuscript and was added in the revised version. 

Even in a feasibility trial, please consider doing a sample size calculation/estimation. 

Response: As mentioned in the manuscript, a sample size calculation was not perform considering the feasibility nature of the study and the pre-experimental design. Sample size calculations for such designs are controversial in the literature. However, we estimated that recruitment of 10 to 15 clinicians and 30 to 45 patients was sufficient to measure feasibility of the intervention, numbers we were able to reach/surpass for this study. 

In the "participants" section, please consider adding criteria related to the characteristics of the LBP (i.e., duration of symptoms [acute, sub-acute or chronic LBP specifying a measurement time] or specific vs non-specific LBP). 

Response: In line with the pragmatic nature of the trial, we purposefully selected broad eligibility criteria to reflect usual practice. Therefore, any patients 18 years or older presenting with a primary complaint of LBP and seeking an initial physiotherapy assessment and willing to provide patient-related outcomes were eligible unless they presented with serious underlying pathology (i.e., red flags). However, we would like to refer the reviewer to Table 3 of the revised manuscript, which now provides much more detailed information about patients’ baseline characteristics. 

"Study setting" section is actually a description of the recruitment. Please consider transferring this section to the "Recruitment" section. 

Response: After careful examination of the manuscript, we agree that the initial formulation could lead to potential confusion between the study setting and recruitment methods. Therefore, we made minor modifications that we believe clarify the distinction between the study setting and participants’ recruitment. 

Line 141: "Clinic managers informed the clinicians working at the different sites." Please explain what the clinic managers informed the clinicians about. 

Response: We would like to thank the reviewer for this observation as we realize that this passage is incomplete and may lead to confusion. Therefore, we provided further details as to what the clinic managers informed the clinicians about in the revised manuscript. Please see the Recruitment subsection in the Methods section of the revised manuscript. 

Is a 1-day workshop enough to provide the clinicians the knowledge and skills they need to achieve the 4 objectives? In my opinion, this training should be carried out in greater detail. Also, authors should consider assessing the competence of clinicians managing the PDDM model after the training process in order to ensure the correct understanding and use of the tool. 

Response: We thank the reviewer for this interesting suggestion. First of all, as mentioned in the study, additional resources were developed and given to the clinicians to support their integration of the model into clinical practice. Second, we completely agree with the reviewer’s comment about assessing the competence of clinicians managing the PDDM model after the training. In fact, we are already planning to add a formal knowledge and skills assessment following the training on the PDDM in a future trial. This point is also clearly stated in the original manuscript in the Discussion section under the subsection Recommendations for a future larger scale trial 

Both applicability of the PDDM model and the feasibility of participants recruitment are not rigorously described in the "Data analysis" section. How the authors have analyzed those outcome measures? 

Response: After reviewing the data analysis section of the original manuscript, we believe that the analyses are adequately described since a detailed description for each outcome is presented in their respective subsection in the Methods section of the manuscript. For these outcomes, a descriptive analysis was used. Nevertheless, we added a brief explanation in the revised manuscript in the Data analysis subsection. 

There is a lack of information about the operation of the model. Please consider describing how the PDDM model classifies patients based on the five domains and what intervention strategies the PDDM suggests based on each category of the model. 

Response: This is indeed an important aspect that needs more clarification. We would like to refer the reviewer to the authors’ response to question 2b from the editor as it addresses this specific issue. 

Please review Table 5. It is written "psyschosocial facors" instead of "psychological factors". 

Response: Table 5 has been revised based on the reviewer’s observation. 

Please consider including a control group in order to be able to correctly interpret a pre-post analysis. 

Response: We completely agree with the reviewer’s comment since a two-arm controlled trial is the next step in the validation process of the PDDM to determine its effectiveness. However, as stated in this study, we initially conducted a small pre-experimental design in order to measure applicability and acceptability of the PDDM model and to explore feasibility and preliminary short-term effect outcomes.

---

## [Decision Letter · Decision Letter 1]

6 Jan 2021

Optimizing management of low back pain through the Pain and Disability Drivers Management Model: a feasibility trial

PONE-D-20-22177R1

Dear Dr. Tousignant-Laflamme,

We’re pleased to inform you that your manuscript has been judged scientifically suitable for publication and will be formally accepted for publication once it meets all outstanding technical requirements.

Kind regards,

Walid Kamal Abdelbasset, Ph.D.

Academic Editor

PLOS ONE

Reviewers' comments:

Reviewer's Responses to Questions

**Comments to the Author**

1. If the authors have adequately addressed your comments raised in a previous round of review and you feel that this manuscript is now acceptable for publication, you may indicate that here to bypass the “Comments to the Author” section, enter your conflict of interest statement in the “Confidential to Editor” section, and submit your "Accept" recommendation.

Reviewer #1: All comments have been addressed

Reviewer #2: All comments have been addressed

Reviewer #3: All comments have been addressed

2. Is the manuscript technically sound, and do the data support the conclusions?

Reviewer #1: (No Response)

Reviewer #2: Yes

Reviewer #3: Yes

3. Has the statistical analysis been performed appropriately and rigorously? 

Reviewer #1: (No Response)

Reviewer #2: I Don't Know

Reviewer #3: Yes

4. Have the authors made all data underlying the findings in their manuscript fully available?

Reviewer #1: (No Response)

Reviewer #2: Yes

Reviewer #3: Yes

5. Is the manuscript presented in an intelligible fashion and written in standard English?

Reviewer #1: (No Response)

Reviewer #2: Yes

Reviewer #3: Yes

6. Review Comments to the Author

Reviewer #1: (No Response)

Reviewer #2: (No Response)

Reviewer #3: it is important to address this issue with the perspective of clinicians. This trial will help many professionals working with LBP.

7. PLOS authors have the option to publish the peer review history of their article (what does this mean?). If published, this will include your full peer review and any attached files.

Reviewer #1: No

Reviewer #2: No

Reviewer #3: No

---

## [Editor Report · Acceptance letter]

8 Jan 2021

PONE-D-20-22177R1 

Optimizing management of low back pain through the Pain and Disability Drivers Management Model: a feasibility trial 

Dear Dr. Tousignant-Laflamme:

I'm pleased to inform you that your manuscript has been deemed suitable for publication in PLOS ONE. Congratulations! Your manuscript is now with our production department. 

Kind regards, 

on behalf of

Dr. Walid Kamal Abdelbasset 

Academic Editor

PLOS ONE